# First-order spatial coherence measurements in a thermalized two-dimensional photonic quantum gas

Tobias Damm[1], David Dung[1], Frank Vewinger [1], Martin Weitz[1] & Julian Schmitt [1]

Phase transitions between different states of matter can profoundly modify the order in physical systems, with the emergence of ferromagnetic or topological order constituting important examples. Correlations allow the quantification of the degree of order and the classification of different phases. Here we report measurements of first-order spatial correlations in a harmonically trapped two-dimensional photon gas below, at and above the critical particle number for Bose–Einstein condensation, using interferometric measurements of the emission of a dye-filled optical microcavity. For the uncondensed gas, the transverse coherence decays on a length scale determined by the thermal de Broglie wavelength of the photons, which shows the expected scaling with temperature. At the onset of Bose–Einstein condensation, true long-range order emerges, and we observe quantum statistical effects as the thermal wave packets overlap. The excellent agreement with equilibrium Bose gas theory prompts microcavity photons as promising candidates for studies of critical scaling and universality in optical quantum gases.

---

[1] Institut für Angewandte Physik, Universität Bonn, Wegelerstr. 8, Bonn 53115, Germany. Correspondence and requests for materials should be addressed to M.W. (email: martin.weitz@uni-bonn.de) or to J.S. (email: schmitt@iap.uni-bonn.de)

Different states of matter are universally characterized by the type of order, which is encoded in their correlation properties[1]. A gas of massive particles at high temperatures or low densities, for example, exhibits short-range spatial first-order correlations as inherited by its classical single-particle properties[2–4]. The latter are determined by the thermal de Broglie wave describing a particle of (effective) mass $m$, which is defined as $\lambda_{th} = h / \sqrt{2\pi m k_B T}$, where $T$ denotes the temperature, equaling the de Broglie wavelength of a particle at an average thermal velocity[5]. Quantum statistical effects emerge in a cold and dense gas when the thermal de Broglie wavelength exceeds the mean interparticle spacing. For massive bosonic particles, as atoms with integer spin, as well as for two-dimensional gases of photons or polaritons, Bose–Einstein condensation (BEC) into a macroscopically occupied ground state has been observed[6–13]. Upon reaching BEC, the achieved macroscopic ground-state population enhances the coherence length far beyond the thermal de Broglie wavelength, essentially covering the complete sample size[3, 5]. This has been strikingly confirmed both in the thermal and condensed-phase regime for massive particles, as ultracold atomic gases[14–19].

For optical quantum gases as photons and exciton-polaritons, an effective mass is established by tailoring the dispersion relation of photons within optical microcavities[20]. Here, upon condensation long-range order has also been revealed, similarly indicating the spontaneous onset of a new phase[9–12, 21–24]. For such two-dimensional systems at high phase-space densities, in principle both BEC and Berezinskii–Kosterlitz–Thouless type of phases may occur, where the former is associated with true long-range order and the latter exhibits first-order correlations decaying algebraically in space[25–34]. In previous exciton-polariton experiments, however, the observed transverse coherence below and partly also above the condensation threshold was limited by the finite cavity linewidth[21, 31]. More recently, Marelic et al.[24] reported on evidence for spatial coherence in a harmonically trapped photon gas, without finding quantitative agreement with Bose gas theory for all particle numbers, as attributed to the finite spatial resolution of the used imaging system and the nonequilibrium nature of the studied optical system far above the condensation threshold. Indeed, the genuine thermal de Broglie wavelength of such two-dimensional photonic gases so far has remained elusive.

Here we report on a quantitative study of the first-order spatial correlations of a two-dimensional harmonically trapped photon gas at equilibrium conditions, both in the classical and in the Bose–Einstein condensed phase using optical interferometry. For this, we have developed an experimental platform excelling in high sensitivity and spatial resolution of more than an order of magnitude below the width of the condensate mode. In the uncondensed phase, we directly observe the thermal de Broglie wavelength through the correlation length of the photons, which has both the expected absolute value and the expected temperature scaling. As the photon number is increased, long-range order, as indicated by a significant increase of the transverse coherence length, emerges when the mean distance between the optical wave packets approaches the measured thermal de Broglie wavelength. Our direct look at the coherence properties of the photons from the interferometrically measured in-plane first-order correlations verifies the good applicability of thermodynamic ideal Bose gas theory for the present photonic quantum gas.

## Results

### Preparation and characterizing measurements.
To prepare a two-dimensional photon gas, we use a microcavity setup filled with a liquid dye solution (see the left-hand side of Fig. 1a). Here the photons are confined by two highly reflective curved mirrors spaced by a distance in the wavelength regime. The correspondingly large free spectral range in the microcavity restricts the dye molecules to, in good approximation, emit only into transversal cavity modes belonging to one fixed longitudinal mode number. The photon gas then becomes two-dimensional with the lowest photon energy at $\hbar\omega_{cutoff} \simeq 2.1\,\text{eV}$ for the transversal ground mode, introducing an effective mass of the cavity photons in our experiment. The resulting quadratic optical dispersion, see Eq. (1) below, is the same as for a massive particle[35]. Figure 1b shows the measured energy–momentum relation for the uncondensed two-dimensional photon gas (see Methods section). The data follow a quadratic scaling with the transverse wave vector $k_r$ with the dashed line showing the expected dispersion for an effective photon mass $m_{eff} = \hbar\omega_{cutoff}/c^2 \simeq 7.76(2) \times 10^{-36}\,\text{kg}$, where $c = c_0/n$ denotes the speed of light in the medium with refractive index $n$, as derived from the cavity parameters. The agreement with the experimental data verifies the predictions for a quadratic (non-relativistic) dispersion in the weak coupling regime[36, 37]. Given the non-vanishing effective photon mass, the concept of a thermal de Broglie wavelength can be extended to the cavity photons. Spatially, the mirror curvature induces a harmonic trapping potential of trapping frequency $\Omega$ for the photon gas, yielding a photon energy in the cavity

$$E \simeq m_{eff}c^2 + \frac{(\hbar k_r)^2}{2m_{eff}} + \frac{1}{2}m_{eff}\Omega^2 r^2. \qquad (1)$$

To achieve thermalization, the photons are coupled to a dye solution in the microcavity (Rhodamine 6 G solved in ethylene glycol) by repeated absorption re-emission cycles. The two transverse modal degrees of freedom thermalize to the (internal rovibrational) temperature $T$ of the dye, leading to Bose–Einstein distributed photon energies of order $\sim k_B T$ above the low-energy cutoff provided that the thermalization is sufficiently fast[35, 38]. To generate a photon gas of total particle number $N$ and compensate for optical losses from, for example, mirror transmission, the dye microcavity is pumped by a laser beam (see Methods section). In the presence of the harmonic trapping potential, the two-dimensional photon gas exhibits a phase transition to a Bose–Einstein condensate at non-vanishing temperatures or finite particle numbers, correspondingly. Above a critical particle number of $N_c = \pi^2/3\ (k_B T/(\hbar\Omega))^2 \simeq 94,000$ photons at room temperature ($T = 300\,\text{K}$), with the harmonic trapping frequency $\Omega/(2\pi) \simeq 37\,\text{GHz}$, BEC has been observed[12, 13, 39], and the temporal evolution into equilibrium has been studied[40, 41].

**Interferometric setup and model.** To investigate the coherence properties of the photon gas, we employ the interferometric measurement schematically depicted in Fig. 1a, see also Methods section for details. The emission from the dye microcavity is collimated and after passing a polarizer to lift the polarization degeneracy sent to a Michelson interferometer with a movable cat-eye retroreflector replacing one of the mirrors[9, 31]. The plane mirror in the reference arm is slightly tilted leading to a fringe-type interference pattern, which we read out with a camera, see Fig. 1c for an example. As the retroreflector inverts the image, each point $r = (x,y)$ in the camera plane corresponds to the interference of fields of the cavity emission at two points at transverse positions $r$ (reference path) and $-r$ (cat-eye path) prior to entering the interferometer. At the detector, we expect an

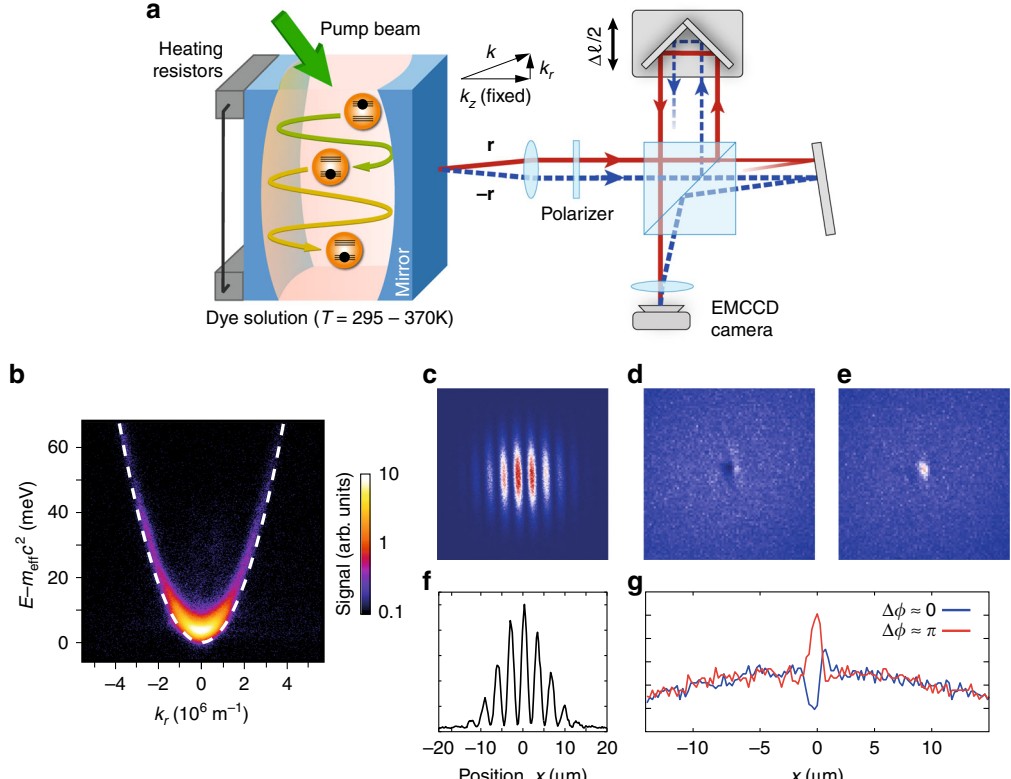

**Fig. 1** Experimental scheme. **a** The dye-filled optical microcavity confines photons between two highly reflecting mirrors spaced in the wavelength regime, where they thermalize to the temperature of the dye solution by absorption re-emission processes. The ambient temperature of the apparatus can be varied between 297 K (room temperature) and 370 K. The Michelson interferometer shown on the *right* is used to investigate the transverse and longitudinal coherence of the photon gas. The sketched axial mirroring of transverse coordinates illustrates the actual point mirroring in the image plane done by the cat-eye retroreflector. **b** Measured dispersion of the dye microcavity emission, showing photon energies vs. the transverse wave vector $k_r$ (derived from the angle of emission). The data were recorded for the uncondensed gas. **c–g** Camera images of the radiation transmitted by the interferometer in the condensed (**c**) and in the uncondensed (**d**, **e**) phase regime, with $N/N_c \approx 1.76$ and $N/N_c \approx 10^{-4}$, respectively. In the uncondensed phase, the transverse coherence length of the microcavity emission is so short that it amounts to less than a fringe width. The two camera images correspond to two different relative path lengths in the Michelson interferometer. The image size corresponds to 40 μm×40 μm (**c**) and 20 μm×20 μm (**d**, **e**), respectively. The *bottom* plots (**f**) and (**g**) give cuts through the center of the corresponding fringe patterns (**c**) and (**d**, **e**), respectively

interference pattern of the form

$$I_{\mathrm{d}}(\mathbf{r}) = \frac{1}{4}\left[I(\mathbf{r}) + I(-\mathbf{r}) + 2\sqrt{I(\mathbf{r})I(-\mathbf{r})}\cos(\Delta\phi)\big|g^{(1)}(\mathbf{r},-\mathbf{r};\tau)\big|\right],$$

(2)

where $I(\mathbf{r})$ corresponds to the intensity distribution of the cavity emission at transverse position $\mathbf{r}$. Further, $\Delta\phi = \arg[g^{(1)}(\mathbf{r},-\mathbf{r};\tau)]$ and $\tau = \Delta\ell/c_0$ denotes the time delay accumulated between arms due to a path length difference $\Delta\ell$. The normalized first-order spatial correlation is defined as[42]

$$g^{(1)}(\mathbf{r},-\mathbf{r};\tau) = \frac{\langle \mathbf{E}^+(\mathbf{r},t)\mathbf{E}(-\mathbf{r},t+\tau)\rangle}{\sqrt{\langle|\mathbf{E}(\mathbf{r},t)|^2\rangle}\sqrt{\langle|\mathbf{E}(-\mathbf{r},t+\tau)|^2\rangle}},$$

(3)

where $\mathbf{E}(\mathbf{r},t)$ denotes the quantized electric field operator at transverse position $\mathbf{r}$ and time $t$ and brackets account for (ensemble) averaging under stationary statistics (see Supplementary Notes 1–4). Experimentally, we use the fringe visibility $V = (I_{\max}-I_{\min})/(I_{\max}+I_{\min})$ as a measure for the correlation function, where $I_{\max}$ ($I_{\min}$) denote maximum (minimum) intensities of the fringe pattern around the path difference $\Delta\ell$.

The visibility is related to the first-order coherence following

$$V(\mathbf{r},\tau) = \frac{2\sqrt{I(\mathbf{r})I(-\mathbf{r})}}{I(\mathbf{r})+I(-\mathbf{r})}\big|g^{(1)}(\mathbf{r},-\mathbf{r};\tau)\big|.$$

(4)

Correspondingly, when varying $\tau$ by changing the longitudinal position of the cat-eye retroreflector with a piezo-driven stepper motor translator, we can extract both longitudinal and transverse coherence properties of the emission from the dye microcavity. Intuitively, the interference signal at each point $\mathbf{r}$ corresponds to the interference expected from a Young's double slit experiment[21, 22] with a slit separation of $2|\mathbf{r}|$, as seen when inspecting the optical paths in Fig. 1a. Figure 1c, f show spatial interference fringes for a fixed path length difference (near $\Delta\ell = 0$) for the case of a photon Bose–Einstein condensate, while Fig. 1d, e, g show images along with a cut along their x axis at $y = 0$ recorded far below the threshold for condensation. For the latter, two path length differences have been selected that correspond to a phase shift of $\pi$. The images indicate the large difference in the transverse coherence length for the case above and below threshold.

We find the expected correlation function Eq. (3) for the case of the two-dimensional photon gas in a harmonic trap similar to earlier work[4, 43]. Briefly, to find $g^{(1)}(\mathbf{r},-\mathbf{r};\tau)$, we expand the electric field operators in eigenfunctions of the harmonic

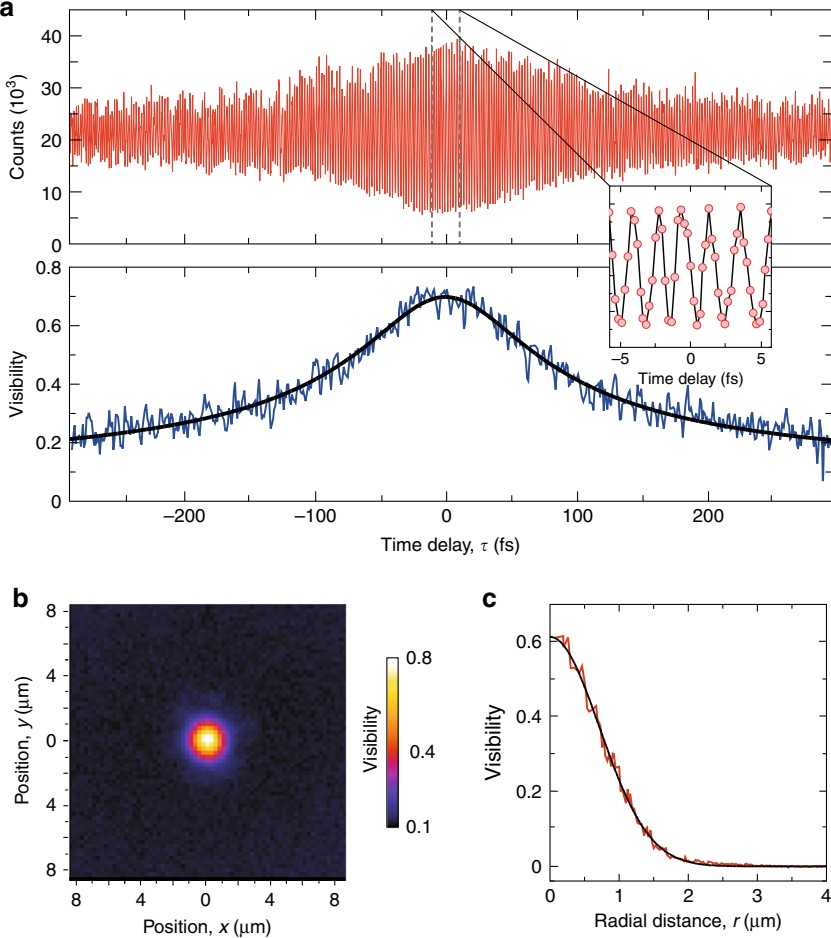

**Fig. 2** Temporal and spatial correlations of thermal photon gas. **a** The *top* panel shows the variation of the camera signal detected by the camera pixel closest to **r** = 0 in the image plane as a function of the time delay due to different interferometer path lengths for a thermal photon gas ($N \ll N_c$). The inset gives a zoomed view into the central fringes. The *bottom plot* shows the corresponding variation of the fringe visibility along with a fit to $|g^{(1)}(\tau)|$ (*solid*). **b** Map of the observed fringe visibility (raw data) vs. transverse position in the camera plane. **c** Offset corrected fringe visibility vs. radial distance from the center (*red*), along with a Gaussian fit (*black*). After correcting for the measured resolution of the imaging system, the curve directly maps the extent of the thermal de Broglie wave packets. ($T = 307$ K, $N/N_c \simeq 10^{-4}$)

oscillator and assume that the photon gas is in thermal equilibrium at temperature $T$. The resulting equations can then be solved numerically, where we use the eigenfunctions of the 700 lowest harmonic oscillator modes; details are summarized in the Supplementary Information. Far below threshold, the distribution function is well described by a Boltzmann distribution, which allows for an analytic solution of the spatial correlation function at equal times, $g^{(1)}(\mathbf{r}, -\mathbf{r}; 0) = \exp(-4\pi|\mathbf{r}|^2/\lambda_{th}^2)$. Correspondingly, a measurement of the spatial correlation length, defined as $\ell_c = \lambda_{th}\sqrt{\ln 2/(4\pi)}$ (full width half maximum (FWHM)), allows to directly determine the thermal de Broglie wavelength. In our numerical calculations, we account for the noise floor of the used camera, which reduces the measured interference contrast especially in regions with low intensities.

**Temporal and transverse coherence**. First, we have studied the temporal coherence by scanning the longitudinal position of the cat-eye retroreflector and studying the interference pattern near **r** = 0. A corresponding temporal fringe pattern is displayed in Fig. 2a (*top*), showing the signal detected by one camera pixel vs. the temporal delay due to path length difference of the Michelson interferometer. The bottom panel shows the corresponding variation of the fringe visibility. The visibility reduces

for large path length differences, and from the decay we find a coherence time in the uncondensed phase of $\tau_c \simeq 129(6)$ fs at a temperature of $T = 297$ K. This value exceeds the expected coherence time of $\sqrt{3}\hbar/(k_B T) \simeq 44$ fs, which we attribute to the finite imaging resolution of our setup (see Supplementary Note 5). Indeed, when including an averaging over an area given by the point spread function in the numerical calculations, the results agree much better with the measured data, demonstrating qualitatively the strong influence of finite spatial resolution on measured temporal coherence data. In accordance with theory predictions, the first-order correlation time of the uncondensed photon gas is more than four orders of magnitude shorter than values observed in the Bose–Einstein condensed phase with heterodyne measurements[23] and slightly smaller than other reported results[24].

To study the transverse coherence, Fig. 2b gives the observed visibilities of the central fringes ($\Delta\tau \simeq 0$ in Fig. 2a) vs. transverse position, directly providing a spatial map of the coherence of the thermal photon gas. To extract the coherence length, we subtract a visibility offset given by the noise characteristics of the camera (see Supplementary Note 6) and radially average the visibility. Figure 2c gives the corresponding variation vs. transverse distance from the origin. The visible rapid decay with increasing distance is understood in terms of the limited transverse coherence of the

thermal photon gas. After correcting for the spatial resolution of the imaging system, measured using a SNOM-fiber of < 200 nm aperture diameter placed in the cavity plane (see Methods section), we can readily determine the spatial correlation length of the two-dimensional, harmonically confined gas. From this, we extract the corresponding thermal de Broglie wavelength $\lambda_{th} = 1.48(1)\,\mu m$. This agrees well with theoretical value of $1.482(2)\,\mu m$, obtained by using the above-quoted effective photon mass and its uncertainty in the formula for the thermal de Broglie wavelength. In contrast to a determination of the de Broglie wavelength by momentum-resolved emission spectra[31], we here directly observe the spatial coherence of the photon wave packet.

To test for the expected temperature scaling of the thermal de Broglie wavelength, the cavity was heated using two electric heaters placed on the side of the cavity mirrors. This allows us to tune the temperature by some 70 K, as at higher temperatures the solvent starts to noticeably evaporate. Figure 3 shows the variation of the experimentally determined thermal de Broglie wavelength with temperature of the dye microcavity. With increasing temperature, the observed transverse coherence length and correspondingly the extracted de Broglie wavelength shortens. The shown error bars give the size of the statistical uncertainty and the *gray line* the systematic uncertainty due to the correction for the point spread function of the imaging system. A fit to the data yields a variation $T^{-0.51(3)}$, which is in very good agreement with the predicted $1/\sqrt{T}$ scaling.

In a next step, the variation of the transverse coherence of the photon gas with increasing photon number was studied. At the onset of BEC, we expect a sharp increase in coherence length, as is qualitatively already visible from inspecting Fig. 1c, f. To quantify this, we measured spatial maps of the fringe visibility for different ratios of the total photon number $N$ and critical photon number $N_c \simeq 94,000$, see Fig. 4a. The corresponding variation of the fringe visibility with radial distance from the center is given in Fig. 4b, demonstrating good agreement with numerical calculations for corresponding values of $N/N_c$ accounting for the detection noise floor and uncertainties in the measured photon number (*shaded areas*). In this parameter regime, continuous operation of the photon gas is not feasible due to excitation of long-lived triplet states of the dye molecules. We therefore use pump pulses of 600 ns temporal length, which is more than two orders of magnitude above the thermalization timescale[40]. This pulsed operation has not been observed to affect the degree of thermalization, but it leads to a signal-to-noise ratio below that of the measurements of the uncondensed gas (Fig. 2). From our data, we find the onset of condensation at a phase-space density $\tilde{n}_c \lambda_{th}^2 = 3.2(6)$ with the critical central density $\tilde{n}_c$ (see Methods section), see Fig. 4c, which corresponds within the uncertainties to the expected value $\tilde{n}_c \lambda_{th}^2 = \pi^2/3$ at criticality[27, 44]. In the condensed phase, the coherence length strongly increases and soon exceeds the condensate diameter (FWHM, *horizontal solid line* in Fig. 4c). For values $N/N_c \ll 1$, corresponding to the thermal regime, the coherence length as expected approaches $\lambda_{th}\sqrt{\ln 2/(4\pi)}$. When comparing our data to the numerically derived coherence function, we see a good agreement both below and above the condensation threshold when we take the camera characteristics into account (*solid line* in Fig. 4c).

Finally, we investigate the photon gas correlations in more detail in a regime close to the condensation threshold where quantum degeneracy is reached. Figure 5 gives the three measured data sets with lowest photon numbers (from Fig. 4b) in the condensed phase regime, fitted with numerically calculated visibility curves using $N/N_c$ as a fitting parameter. Our previous discussion has identified two distinct regimes: first, we find correlations in the uncondensed photon gas in good agreement with a Gaussian decay on a length scale given by the thermal de

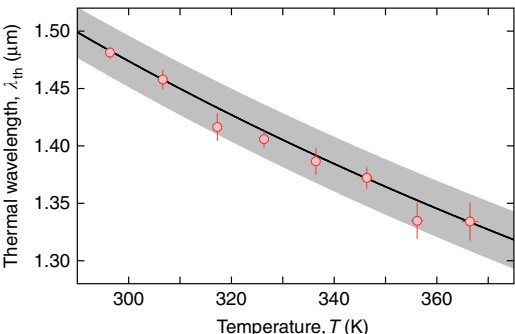

**Fig. 3** Temperature dependence of the thermal de Broglie wavelength. Variation of the determined thermal de Broglie wavelength of the two-dimensional photon gas with the temperature of the dye microcavity apparatus. The data are in good agreement with the theory curve $\lambda_{th} = h/\sqrt{2\pi m_{eff} k_B T}$ (*solid line*), expected for a particle with mass $m_{eff} = \hbar \omega_{cutoff}/c^2$ at temperature $T$. The *shaded area* indicates the systematic uncertainty due to the correction for the measured finite imaging system resolution. The statistical uncertainty (s.d.) is indicated by the *error bars*

Broglie wavelength (Fig. 2), and second, coherence exceeds the ground mode diameter in the Bose–Einstein condensed system (Fig. 4). For the intermediate region, one analytically expects the spatial correlations to consist of a Gaussian thermal contribution $\exp(-4\pi |\mathbf{r}|^2/\lambda_{th}^2)$ at short length scales, as determined by the population in highly excited transverse modes, and an exponentially decaying quasi long-range contribution $\exp(-2|\mathbf{r}|/\xi)$, which results from the macroscopic population of the low-energy states at quantum degeneracy[2, 27]. Here, $\xi$ denotes the correlation length. In the thermodynamic limit, the described bimodal decay of correlations is expected only in the uncondensed phase ($N < N_c$), while upon condensation directly true long-range order, with $\xi \to \infty$, emerges. At photon numbers of order of $10^5$ for the here studied two-dimensional photon gas, however, finite-size effects become important, leading to a softening of the phase transition, which is associated with a continuous gradual increase of the correlation length. Correspondingly, the bimodal behavior is visible in our experimental and numerical data also for $N > N_c$, as seen in Fig. 5. The corresponding expected population of low-energy states is given in the spectral photon distributions (inset).

From an exponential fit to the experimental data in Fig. 5 between 2 and 6 $\mu$m radial distance (*dashed lines*), we obtain a correlation length $\xi = 7.4(2)\,\mu m$ for a total photon number of 95,000 ($N/N_c = 1.01$) and $\xi = 9.6(5)\,\mu m$ for 96,000 photons ($N/N_c \simeq 1.02$). For both of these measurements, the correlation length is of order of the size of the ground mode given by the oscillator length $\sigma_0 = \sqrt{\hbar/(m\Omega)} = 7.7\,\mu m$. Measurements closer to the condensation threshold of $N_c = 94,000$ in the harmonically trapped case would require a photon number precision on the order of 0.1%. For $N \lesssim N_c$, we cannot extract accurate experimental values for $g^{(1)}(|\mathbf{r}|)$ due to limited signal-to-noise ratio in the here required pulsed pumping operational mode, and correspondingly we only show results of our numerical calculations. Nevertheless, our analysis reveals the predicted emergence of (quasi) long-range order in the regime close to the condensation threshold[2, 27].

To conclude, we have determined spatial coherence properties of a two-dimensional photon gas both in the thermal and the Bose–Einstein condensed phase. The high spatial resolution of the interferometric setup allows us to directly image the coherence properties of the photon gas. In the uncondensed regime, our measurements reveal that the extent of the photon wave packets is determined by the thermal de Broglie wavelength. We find

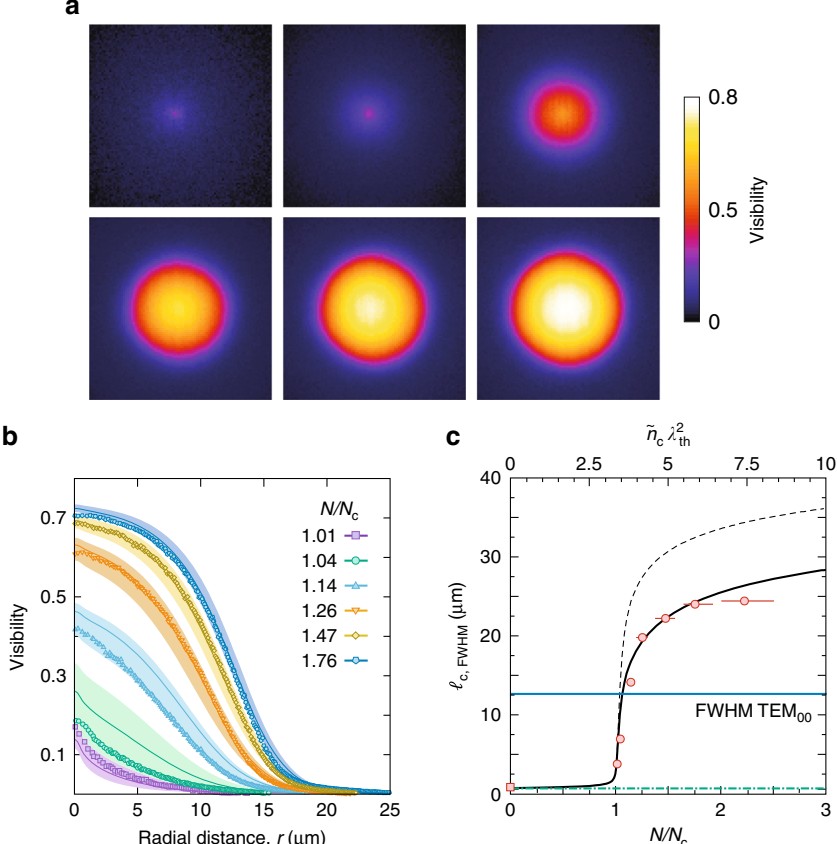

**Fig. 4** Emergence of coherence for a Bose–Einstein condensed photon gas. **a** Maps of the fringe visibility with transverse position for increasing ratios $N/N_c = \{1.01; 1.04; 1.14; 1.26; 1.47; 1.76\}$ showing a strong increase in the coherence, which indicates the onset of long-range order. The size of each image corresponds to 40 μm × 40 μm. **b** Corresponding variation of the fringe visibility (radially averaged) with distance from the center for different total photon numbers. The measured data (*symbols*) is accompanied by numerical simulations (*solid lines*) after taking into account the noise characteristics of the used EMCCD detector and the uncertainty in the experimental determination of the total photon number (*shaded areas*). **c** Measured transverse coherence length, with the dots (*square*) corresponding to data recorded with pulsed (cw) pumping, respectively, vs. $N/N_c$ (*bottom axis*) and phase-space density $\tilde{n}\lambda_{th}^2$ (*top axis*) along with theory for the coherence length with (*dashed line*) and without (*solid*) numerically implementing technical limitations. Far below the threshold to a Bose–Einstein condensate, the transverse coherence length equals $\lambda_{th}\sqrt{\ln 2/4\pi}$ (*dashed-dotted line*), and upon reaching Bose–Einstein condensation the ensemble becomes transversally coherent, with correlation lengths exceeding the 12.7 μm FWHM of the condensate mode (*blue solid line*). The *error bars* represent the uncertainty (s.d.) in the determination of $N/N_c$

excellent agreement both in the absolute value as well as the temperature scaling of the de Broglie wavelength with the expectations. We observe that quantum statistical effects, as indicated by long-range order, emerge when the thermal de Broglie wave packets spatially overlap, a behavior so far only verified for atomic gases. For the condensed phase, we find that the coherence extends over the whole sample. Close to the phase transition, the quantum degenerate gas exhibits thermal and quasi-long-range contributions to the spatial correlations as expected theoretically. To further explore this regime, it would be beneficial to study the photon gas confined in a box potential at large phase-space densities, which would allow a more quantitative comparison with theory predictions due to the absence of BEC in the homogeneous two-dimensional system[17, 27, 45].

For the future, spatially resolved first-order coherence measurements are expected to reveal possible long-lived phase singularities from vortices in thermo-optically or Kerr nonlinearity-induced photon superfluids[46, 47]. Other than atomic condensates, optical quantum gases can be subject to grand canonical statistics[48, 49], which is expected to give rise to unusual dynamics of the quantum fluid. Finally, our setup might be a tool to study critical scaling at the phase transition.

## Methods

**Dye-cavity setup and characterization.** The dye microcavity uses two spherically curved high-reflecting dielectric mirrors (1 m radius), as typically used in cavity ring-down spectroscopy with reflectivities >99.997% in the relevant wavelength regime (530–585 nm). The mirrors are separated by $D_0 = 1.63$ μm, corresponding to four optical wavelengths at 583 nm in the solvent (refractive index $n$(T=297(3) K) = 1.431(3)), causing a large frequency gap between adjacent longitudinal optical modes that is comparable to the emission bandwidth of the dye molecules. Thus the resonator is populated only with photons of a fixed longitudinal mode ($q = 8$) making the photon gas two dimensional. The cavity is filled with Rhodamine 6G dye solved in ethylene glycol (dye concentration $10^{-3}$ mol/l), acting as a heat bath for the photon gas. Collisions of solvent and dye molecules ($10^{-14}$ s timescale) here suppress coherent energy exchange between photons and dye molecules, so that the photon gas is operated in the weak coupling regime[37]. The dye microcavity is pumped with a laser beam of ~ 100 μm diameter at 532 nm, exploiting a minimum in the mirror reflectivity at 43° angle to the optical axis. In the uncondensed regime, pumping is done continuously with a pump power of ~ 1 mW, yielding $N$ = 18(2) intracavity photons. However, as the optical pump power required to reach the critical photon number for condensation favors excessive population of long-lived triplet states and photo bleaching of the organic dye molecules, continuous pumping is rendered unfeasible in the condensed phase. Therefore, using two acousto-optic modulators, the beam is chopped into pulses of 600 ns length with a 50 Hz repetition rate. To allow for a heating of the dye microcavity above room temperature, two electrical power resistors are glued to the cavity mount, allowing for a 12 W thermal output. Further, two thermo-resistive temperature detectors are attached to the cavity mirror substrates from different cavity sides, and the quoted temperatures of the dye microcavity apparatus denote the average reading of these

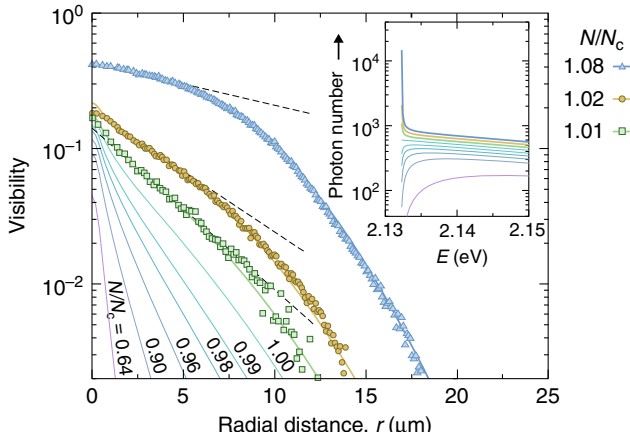

**Fig. 5** Correlations near the condensation threshold. First-order correlations (*symbols*, see also Fig. 4b) close to the critical photon number of $N_c = 94000$ for three values of $N/N_c$, which have been determined by comparison with numerical calculations (*solid lines*). All three data sets follow closely the expected theoretical behavior. For $N/N_c = 1.01$ (*squares*) and 1.02 (*circles*), we observe a short-range contribution to the correlations as attributed to Gaussian thermal correlations, while between 2 and 6 µm the correlations reveal an exponential decay resulting from quantum degeneracy of low-energy states (see inset). For radii beyond the oscillator length $\sigma_0 = 7.7$ µm, correlations decay due to the finite condensate width, as visible in the data for $N/N_c = 1.08$ (*triangles*). Exponential fits (*dashed lines*) to the data yield correlation lengths $\xi = \{7.4(2); 9.6(5); 21.6(29)\}$ µm for $N/N_c = \{1.01; 1.02; 1.08\}$. For the uncondensed phase ($N/N_c < 1$), where data acquisition is prohibited by insufficient SNR of the imaging system, we show numerical calculations to illustrate the behavior of the photon gas correlations (*solid lines*). Inset: Theory spectra of the photon distribution (vertically shifted) for corresponding values of $N/N_c$ show the balanced population of the lowest-energy modes near the condensation point

detectors with the error bar (Fig. 3), reflecting the difference between the two readings.

The imaged dispersion shown in Fig. 1b corresponds to a momentum-resolved spectrum of the cavity emission. For this, only the central ~ 30 µm (diameter) of the cavity emission is transmitted through an iris aperture in the image plane, which effectively reduces the relative intensity contribution of the higher energy modes. The light subsequently passes a narrow slit in the momentum plane, before it is sent onto an optical grating (2400 lines/mm) and imaged onto the camera.

**Interferometer setup**. To characterize the first-order coherence of the photon gas, the emission on one side of the microcavity is collimated with a long working distance objective (Mitutoyo M Plan Apo 10×) and sent onto a Michelson-type interferometer. After passing a polarizer, the beam here is split up equally into two partial beams with a non-polarizing beam splitter, which are reflected by a plane mirror and a hollow cat-eye retroreflector, respectively. The latter is mounted on a piezo-driven stepper motor translator allowing for a steady variation of the path length difference of the interfering paths. The beams are recombined in the beam splitter following the usual Michelson interferometer arrangement and then imaged on an EMCCD camera. The hollow mirror cat-eye retroreflector inverts the two transversal spatial coordinates (transverse with respect to the optical axis) and is illuminated off-center to bypass imperfections at the internal contacting edges of the device. Correspondingly, a slight tilt of the plane mirror back reflector is needed to match the two beams in the image plane. To observe interference fringes, we sample images for different delay times, typically scanning the cat-eye retroreflector over a longitudinal distance of roughly 90 µm, corresponding to a variation of the total time delay of 180 µm/$c_0 \approx 600$ fs. During such a scan, 2500 images are acquired, corresponding to time steps in the path difference near 0.2 fs.

The obtained fringe visibility recorded for each of the camera pixels allows to generate a two-dimensional map of the correlation function $g^{(1)}(\mathbf{r}, -\mathbf{r}; \tau)$, where $\tau = \Delta\ell/c_0$, denotes the time delay accumulated due to the path length difference $\Delta\ell$. The transverse coherence length of the photon gas (Figs. 2b, c and 4) is obtained by evaluating fringe data with a path length difference much below the longitudinal coherence length, yielding effectively $g^{(1)}(\mathbf{r}, -\mathbf{r}; \tau) \equiv g^{(1)}(\mathbf{r}, -\mathbf{r})$. The reduced maximal visibility in Fig. 2c is attributed to the influence of detector noise floor (see Supplementary Note 6). As the transverse coherence length of the

uncondensed photon gas is of the same order of magnitude as the imaging resolution of the objective used to collimate the microcavity emission, the point spread function of the whole imaging system was carefully determined in preliminary measurements. To simulate the emission of a point source, the < 200 nm diameter aperture of a SNOM fiber tip (LovaLite EM50 SMF28) was placed in the emission plane of the cavity, the latter replaced by a non-reflecting cavity dummy of same shape and size, using a drop of ethylene glycol for index matching purposes. The resulting image of the fiber emission (utilizing a dye laser at 583 nm) is used to characterize the complete imaging system. Both the spatial decay of the experimentally observed visibility data vs. transverse position for a thermal photon gas, see Fig. 2c, as well as the imaging system point spread function can be well approximated by Gaussian curves, with the latter exhibiting a width of $\sigma_{PSF} \simeq 0.658(2)$ µm. A deconvolution of the measured Gaussian correlation signal of width $\sigma$ with the point spread function can thus readily be performed, and the true correlation length is obtained by $\sqrt{\sigma^2 - \sigma_{PSF}^2}$. The finite spatial resolution of the imaging system also implies that the temporal interference signal shown in Fig. 2a corresponds to data averaged over a spatial area corresponding to the imaging resolution noted above, which is larger than the ~200 nm × 200 nm size of one camera pixel in the imaging plane.

**Measurement details**. The measured absolute value of the coherence time of $\tau_c \simeq 129(6)$ fs at $T = 297$ K temperature corresponds to $\tau_c = \kappa\hbar/(k_B T)$ with $\kappa = 4.90(3)$, which is larger than the expected $\kappa = \sqrt{3}$ for the case of perfect imaging resolution (see Supplementary Note 3). This results from the finite spatial resolution, which mixes spatial correlations at different positions to yield the observed temporal correlation data (see Supplementary Note 5). The data shown in Figs. 2 and 3 were recorded at typical photon numbers in the dye microcavity apparatus of $N = 18(2)$, which is more than three orders of magnitude below the critical photon number $N_c = 94,000$.

The experimental data shown in Fig. 4 investigates the condensed phase regime of the photon gas with condensate fractions ranging from 1% up to 55%, corresponding to ratios $N/N_c$ of up to 2.2(3). At the used total photon numbers, we find no evidence for a spatial broadening of the condensate, so that thermo-optic interaction effects are considered to be small. In comparison with ref.[12], we attribute this behaviour to the here used longer spacing between successive pump pulses. The quoted values for $N/N_c$ in Fig. 4 are determined from a spectroscopic measurement of the cavity emission using a spectrometer in 4*f* configuration, equipped with a motorized slit for wavelength selection and a photomultiplier. To allow for a comparison of the numerically obtained visibilities with the results extracted from the experimental data, we have accounted for the noise characteristics of the used detector, with an intrinsic noise floor present in the taken images. This effect reduces the resolvable visibility in the outer parts of the wings of the Gaussian condensate mode as the mean noise floor here becomes comparable to the expected cavity emission intensity, which reduces the obtained value for the coherence length. For details and a more elaborated discussion of the performed numerical simulations on correlations in a thermalized two-dimensional photon gas confined in a harmonic trap, see Supplementary Notes 1–4.

The absolute photon number at criticality is found by evaluating spatial images of the emission recorded with a calibrated EMCCD camera. By considering the cavity characteristics (mirror transmission, round trip time, energy cutoff and pulse length), the transmission of all optical elements in the beam path and the detector characteristics (quantum efficiency, gain, electrons per count and detected areal fraction of the emission) with respective uncertainties, we experimentally find the critical intracavity particle number $N_c = 90(18) \times 10^3$ to be in good agreement with the theoretically expected value. We determine the critical phase-space density $\bar{n}_c \lambda_{th}^2 = 3.2(6)$ from the central density $\bar{n} = N/[\pi(\Delta r)^2]$, see refs.[27, 44], with the thermal cloud radius $\Delta r = \sqrt{2k_B T/(m_{eff}\Omega^2)}$. Note that the expected critical phase-space density $\bar{n}_c \lambda_{th}^2 = \pi^2/3 \approx 3.3$ differs from the value $\zeta(2) = \pi^2/6$ of the Riemann zeta function $\zeta$ for a two-dimensional system in a harmonic potential due to the intrinsic twofold polarization degeneracy of the photonic system.

**Data availability**. The data that support the findings of this study are available from the authors upon reasonable request.

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

## Acknowledgements

This work has been supported by the DFG (CRC 185) and the ERC (Inpec).

## Author contributions

T.D. carried out the experimental measurements, with contributions from J.S. Theoretical modeling was performed by J.S. All authors analyzed and discussed the results. T.D., J.S., F.V. and M.W. wrote the manuscript.

## Additional information

**Competing interests:** The authors declare no competing financial interests.

