## [Peer Review File · Nature Communications]

Reviewers' comments:

Reviewer #1 (Remarks to the Author):

This paper presents measurements of first order coherence of thermalised and condensed photon gases. The coherence measurements are used to determine the thermal de Broglie wavelength rather precisely. The data is of a very high quality, and the match to thermal equilibrium theory is very tight. As such, the paper does not present any surprises, but it does set a new standard of data quality for the field of photon and polariton condensates, and should prove to be a stepping stone to investigation of the important problem of critical scaling near the condensation phase transition.

The conclusions of the paper are well justified by the data and the accompanying theoretical analysis (when taking into account the supplementary material). In my opinion, the manuscript is acceptable for Nature Communications in very nearly its present form, with only a few changes necessary. Those changes are: consistency of definitions (imaging resolution and effective mass) and units, and a discussion of the effect of polarization on fringe visibility. I will also suggest a couple of optional points that you could choose to include to strengthen the article and potentially improve its impact.

The discussion of the dispersion relation seems a little out of place in the abstract, and Figure 1b seems to detract from the main points in the paper unless the relationship between the dispersion relation being quadratic and the concept of the thermal de Broglie wavelength is a key point. If so, this should be clarified early in the article. I agree with the literature review conclusion, that the thermal de Broglie wavelength of a photonic gas has never yet been directly measured. The measurement of the dispersion relation is very clear, but what is the dashed line? It looks like the dispersion for a massive particle, but the mass is not defined. I will also note that Roumpos et al, PNAS (2012) fig 4 shows how the dispersion relation relates to the spatial and temporal coherence.

It is remarkable that you have managed to resolve such a small length so precisely through an interferometer. There does seem to be an inconsistency in your statement of the imaging resolution: 0.77 microns in the main text, 0.93 micron in the supplementary material. Please correct this.

You use both electron-Volts and wavelength, and define an angular frequency to describe the light's energy: please stick to one convention throughout the paper.

There seems to be some discrepancy between the formula for the photon mass and the measured and stated de Broglie wavelength, if I use $c = 3 \times 10^8$ m/s. Does the symbol "c" stand for the speed of light in vacuum or in the medium? Please define this carefully in your article.

Why do you stop at 370 Kelvin, and not go to higher temperatures?

Equation (2) is for stationary statistics, which should be stated in the main text. In fact, a summary of the discussion from the supplementary material should be in the main text. Purely based on the main text, the reader is left with no idea what model fits the data, i.e. what physical processes actually govern the observed behaviour (which turn out to be just thermal equilibrium, and finite resolution and detector noise). For example, on line 109, the formula for transverse coherence is given, with no explanation or citation.

This article has a certain amount of overlap of content with Reference [7]. The authors should make clear exactly what they have discovered that is not found within [7]. A very similar theoretical analysis (in the supplementary material) is also found in [7], and it's references 48 to 50, and so could be effectively summarised with a few good references.

The authors state that there is a twofold polarization degeneracy, which affects the critical phase space density. One would expect that the polarization of the light would also affect the visibility of fringes. It might be appropriate to define the coherence with yet more labels, for the polarization modes i and j : $g(1)(r,r',\tau, i,j)$. Please discuss this in your main text as well as the supplementary material.

There is seems to be a lack of a factor $1/2$ on line 301. I thought that the virial theorem states that for a harmonic trap, kinetic energy = potential energy = $kT/2$.

It might be appropriate to cite Guarrera et al, PRL 2011 (and 2012) for temporal correlations in trapped bose gases. Note that the second order correlation function is, at thermal equilibrium, directly related to the first order coherence: $g(2) = 1+|g(1)|^2$.

Refs 20 and 21 seem to be in the wrong order, and Ref 18 has a proper journal reference available (PRA, 2016).

For the caption of figure 2, please state that "thermal gas" means $N/N_c \ll 1$. It took me a while to work that out.

What is the red line in Figure 1c, bottom left panel?

Why does figure 2b ($N \ll N_c$) have visibility near unity, but figure 4a ($N/N_c = 1.01$) has no more than about 0.3? Surely the visibility should just increase with increasing particle number.

In the supplementary material, section 2, "for the sake of simplicity" you have thrown away something interesting, namely correlations at different times not at $r=r'$. Correlations over time for two different positions can be seen to be non-trivial from the Roumpos et al, PNAS, 2012, showing how correlations propagate. Not only could your calculations presumably be extended to include these correlations, also your data presumably can be re-analysed to extract them. While including this analysis would not be necessary for publication in Nature Communications (in my opinion), it would enhance the importance and interest in your article if you found propagating correlations.

What is the origin of the numerical coefficients in equation 29 of the supplementary material? Also, in the supplementary material, "Sakurai" is misspelt.

Reviewer #2 (Remarks to the Author):

This is a nice experiment in many ways, basically a redo of the in-plane ("transverse") coherence measurements of polariton condensates, esp. Ref. 14, for the purely photon condensate. This has the additional feature that the photons are well thermalized, while in many polariton condensates the gas is not fully thermalized.

However, this paper is strangely presented, and feels in some ways like a work in progress rather than a fully developed work.

First, the title and abstract present the main point as measuring the thermal DeBroglie wavelength. But this is not much new. The DeBroglie wavelength is a single-particle property that does not depend on many-body correlations. It can easily be determined (and has been determined) in polariton and photon gas experiments by simple measuring the average k -vector in k -resolved emission spectra.

Making this paper mainly about "we can deduce the DeBroglie wavelength through a much more difficult and involuted process" is poor marketing.

This paper is really about the in-plane correlation measurements, and as such should be titled simply "First-order coherence measurements in a 2D..." Then at low density the interesting story is that a theoretical prediction (line 109 on page 5) is confirmed. When I first read the paper it seemed this formula was just pulled out of the air with no justification or citation. But it turns out it is extensively justified in the SI file. The authors need to make this connection clearly by referring to the SI file and giving a brief description of the theory presented there.

More interesting to me is the measurement of the correlation/coherence length in the BEC regime. Nice results for this are given, and are compared to theory in Figure 4c. But once again, on first reading I could not find a citation, discussion, or formula for the dotted and solid curves in Figure 4c (or the curves in 4b). The connection to the SI file (i.e. directly citing the SI file) and a brief summary of the theoretical approach needs to be in the main text.

My conclusion is that this is indeed an interesting paper, but needs major revision. The authors have made it seem as though it is a breakthrough to be able to measure the thermal wavelength (a single-particle property) in the real-space regime rather than the k-space regime. Rather, the main story of this paper needs to be the confirmation of the theory for correlations, which is presently buried in the SI file.

Reviewer #3 (Remarks to the Author):

The manuscript by Damm et al. reports an experimental study of the coherence properties of a Bose gas of photons in a dye-filled optical microcavity. Using a quite standard optical set-up for detection of spatio-temporal coherence, the authors characterize the behaviour of the first-order g_1 coherence function in the two extreme regimes of photon number well below and above the condensation critical point.

I have mixed feelings about this manuscript.

The general idea of this measurement is a quite standard one, and very similar experiments have been carried out with other related systems, in particular polaritons. However, I have to admit that the quality of the experimental data is much cleaner than the one of most polariton experiments which are typically plagued by sample disorder. Here the very smooth data curves make the conclusions very convincing. Some results are novel and typical of photon BEC's, e.g. the measurement of the system-temperature-dependence of the de Broglie wavelength. The overall high quality of the data reinforces the feeling that photon BEC are indeed an excellent work-horse for studies of more sophisticated aspects of long-range phase coherence.

As a most serious flaw of the manuscript, I am afraid that the study is still incomplete as it does not cover all possible regimes around the BEC critical point. From what I remember of the general theory of BEC (see e.g. the 1999-2000 Cohen-Tannoudji's lectures available on the College de France website <http://www.phys.ens.fr/~cct/college-de-france/1999-00/1999-00.htm>), the g_1 has three main regimes as a function of particle density: well below condensation (where it matches the de Broglie wavelength), above condensation (where coherence extends over the whole sample) and an intermediate regime when the system is quantum degenerate and the coherence length is far longer than the de Broglie wavelength, but the system is not yet condensed. This last regime is never explicitly mentioned in this manuscript, which at several points rather gives the potentially misleading impression that BEC occurs as soon as quantum degeneracy is reached. A trace of this behaviour is

anyway visible in fig.4c as the theoretical solid curve start deviating from the classical thermal value already well before the $N/N_c=1$ critical point. For the manuscript to be complete, I think that some discussion of this regime is essential, with some experimental data showing the increased coherence length as an effect of quantum degeneracy before condensation.

In summary, my feeling is that the manuscript is interesting and promising, but it still requires additional data and additional discussion to be complete and ready for publication. In its present form I can not recommend it for publication, as it might have a misleading effect on the community. If the authors agree in performing these additional experiments and correspondingly updating the text, I am looking forward to see a revised version of the manuscript for further review.

Reply to referee and list of changes

We are pleased about our manuscript being in general very well received and appreciated by all three referees and thank for their insightful comments. We have modified the manuscript in light of their remarks, as pointed out by our responses (indented).

Reply to Referee #1:

Referee: This paper presents measurements of first order coherence of thermalised and condensed photon gases. The coherence measurements are used to determine the thermal de Broglie wavelength rather precisely. The data is of a very high quality, and the match to thermal equilibrium theory is very tight. As such, the paper does not present any surprises, but it does set a new standard of data quality for the field of photon and polariton condensates, and should prove to be a stepping stone to investigation of the important problem of critical scaling near the condensation phase transition.

The conclusions of the paper are well justified by the data and the accompanying theoretical analysis (when taking into account the supplementary material). In my opinion, the manuscript is acceptable for Nature Communications in very nearly its present form, with only a few changes necessary. Those changes are: consistency of definitions (imaging resolution and effective mass) and units, and a discussion of the effect of polarization on fringe visibility. I will also suggest a couple of optional points that you could choose to include to strengthen the article and potentially improve its impact.

The discussion of the dispersion relation seems a little out of place in the abstract, and Figure 1b seems to detract from the main points in the paper unless the relationship between the dispersion relation being quadratic and the concept of the thermal de Broglie wavelength is a key point. If so, this should be clarified early in the article. I agree with the literature review conclusion, that the thermal de Broglie wavelength of a photonic gas has never yet been directly measured. The measurement of the dispersion relation is very clear, but what is the dashed line? It looks like the dispersion for a massive particle, but the mass is not defined. I will also note that Roumpos et al, PNAS (2012) fig 4 shows how the dispersion relation relates to the spatial and temporal coherence.

1. Our reply: We agree with the Referee that the discussion of the dispersion relation in the abstract shifted the point of interest in disfavor of our actual findings. We rearranged the abstract to emphasize that the measurement of the correlation

properties is central to this study. However, we point out that the quadratic dispersion relation is directly linked to the used thermal de Broglie wavelength of a massive particle and thus a key aspect of two-dimensional, harmonically trapped photon gas. See e.g. Z. Yan, EJP (2000) for a generalized definition of the thermal wavelength and its direct dependence on dimension and dispersion. We clarified the appropriate part in the main text. Accordingly, we added a description of the dashed line shown in Fig. 1b to the figure caption and changed the energy scale to read “ $E - m_{\text{eff}}c^2$ ”.

We like to comment on the referee’s note of the relation between the dispersion relation and spatial coherence: We are aware that in general the Fourier transform of the measured spectrum in k-space is equivalent to a direct in-plane measurement using interferometry. However, we preferred to directly measure the in-plane correlations because Fourier transformation of the momentum spectrum provides no technical advantage for our harmonically trapped system. Even in exciton-polariton systems, as stated by Roumpos et al. in Ref. [31], “the direct measurement of $g^{(1)}(x, -x; t)$ is the only way to reliably extract λ_{eff} ”.

It is remarkable that you have managed to resolve such a small length so precisely through an interferometer. There does seem to be an inconsistency in your statement of the imaging resolution: 0.77 microns in the main text, 0.93 micron in the supplementary material. Please correct this.

2. The extent of the point spread function in the main text and Supplementary information was given for different definitions of the width of the Gaussian distribution.

We now use the standard deviation σ_{PSF} to quantify the width consistently throughout the manuscript.

You use both electron-Volts and wavelength, and define an angular frequency to describe the light’s energy: please stick to one convention throughout the paper.

3. To give an intuitive access to energy scales we use electron volts in the main text. For the same reason, we use wavelengths in the methods section to describe the technical details of the setup located in the visible regime.

There seems to be some discrepancy between the formula for the photon mass and the measured and stated de Broglie wavelength, if I use $c = 3 \times 10^8$ m/s. Does the symbol “c”

stand for the speed of light in vacuum or in the medium? Please define this carefully in your article.

4. The symbol “ c ” denotes the speed of light in the medium, while “ c_0 ” denotes the speed of light in vacuum. We added a clarification of this terminology at the first appearance in the main text. The effective mass m_{eff} used throughout the manuscript is given by $m_{\text{eff}} = \hbar \omega_c (n/c_0)^2 \approx 7.8e^{-36}$ kg, with $n=1.43$ as the refractive index of the dye solvent (ethylene glycol).

We added an appropriate definition of the effective photon mass in the main text of the manuscript.

Why do you stop at 370 Kelvin, and not go to higher temperatures?

5. Although the boiling point of ethylene glycol is at 470 K, we encountered technical issues for temperatures higher than the reported 370 K. The accelerated evaporation of the solvent inside our cavity introduced mechanical vibrations of amplitudes which could not be damped by an active stabilization. Thus, a precise measurement of the spatial correlations becomes unfeasible.

We clarified this in the corresponding paragraph.

Equation (2) is for stationary statistics, which should be stated in the main text. In fact, a summary of the discussion from the supplementary material should be in the main text. Purely based on the main text, the reader is left with no idea what model fits the data, i.e. what physical processes actually govern the observed behaviour (which turn out to be just thermal equilibrium, and finite resolution and detector noise). For example, on line 109, the formula for transverse coherence is given, with no explanation or citation.

6. The brackets $\langle \dots \rangle$ in Eq. (3) (former Eq. (2)) denote an ensemble averaging over different realizations. Thus, the explicit time dependency is replaced by a time delay. We further agree with the Referee that parts of the necessary information remained in the Supplementary information without adequate reference in the main text.

In the revised manuscript, we added material from the Supplementary information. The stated formulas in the main text can now be well understood without the explicit derivation still presented in the Supplementary information.

This article has a certain amount of overlap of content with Reference [7]. The authors should make clear exactly what they have discovered that is not found within [7]. A very similar theoretical analysis (in the supplementary material) is also found in [7], and its references 48 to 50, and so could be effectively summarised with a few good references.

7. In the work of Marelic et al. (Reference [24]; former Reference [7]) the spatio-temporal first-order coherence of a dye-photon-system under nonequilibrium conditions is studied. Hence, their experimental findings differ from those obtained in the near equilibrium system presented in our work, despite of the similarity of the experimental approach. Also, the spatial resolution of our imaging system is higher than the one used in this earlier work.

We clarified this in the introductory part of the main text.

The authors state that there is a twofold polarization degeneracy, which affects the critical phase space density. One would expect that the polarization of the light would also affect the visibility of fringes. It might be appropriate to define the coherence with yet more labels, for the polarization modes i and j : $g(1)(r,r',\tau, i,j)$. Please discuss this in your main text as well as the supplementary material.

8. The two orthogonal polarizations of the electric field can be treated independently in the theoretical model describing the photon gas correlations. The critical phase space density per polarization mode corresponds to the usual value for a two-dimensional, harmonically trapped ideal Bose gas. Besides a factor of two in the critical total particle number to account for both polarizations, we do not expect any polarization-related effects, such as e.g. a reduced fringe visibility. Nevertheless, in the experiment we can see a slight linear polarization of the condensate mode which we attribute to stress-induced birefringence in the cavity mirrors. We therefore filter the emitted light with a polarizer.

We added a comment on the polarization filtering and modified Fig.1a.

There is seems to be a lack of a factor 1/2 on line 301. I thought that the virial theorem states that for a harmonic trap, kinetic energy = potential energy = $kT/2$.

9. Indeed, according to the virial theorem the kinetic as well as potential energy account for $k_B T/2$ each, multiplied by the dimensionality of the system. The two dimensional, harmonically trapped photon gas has four (two kinetic + two potential) degrees of freedom, therefore the previously mentioned equation is fulfilled.

We removed the according sentence to avoid a potential misleading statement.

It might be appropriate to cite Guarrera et al, PRL 2011 (and 2012) for temporal correlations in trapped bose gases. Note that the second order correlation function is, at thermal equilibrium, directly related to the first order coherence: $g(2) = 1+|g(1)|^2$.

10. We thank the referee for suggesting the reference Guarrera et al., and added it to our manuscript (Ref. [18]).

We further would like to comment on the referees note that $g^{(2)}=1+|g^{(1)}|^2$ in thermal equilibrium. This relation is rather connected to thermal number statistics than to thermodynamic equilibrium, which in general are not equivalent. It further assumes statistically independent emitters, a prerequisite no longer fulfilled in the condensed phase of the photon BEC. In earlier work, we investigated the number statistics of the photon condensate, finding statistical regimes with Poissonian ($g^{(2)}(0)=1$) as well as Bose-Einstein-like ($g^{(2)}(0)>1.5$) number fluctuations, as attributed to (grand) canonical statistical ensemble conditions (Refs. [23,47,48]). In all cases, the photon gas was observed to be in thermodynamic equilibrium.

Refs 20 and 21 seem to be in the wrong order, and Ref 18 has a proper journal reference available (PRA, 2016).

11. We reordered the list of references and added the full journal reference for Reference [35] (formerly [18]).

For the caption of figure 2, please state that "thermal gas" means $N/N_c \ll 1$. It took me a while to work that out.

12. We rephrased “thermal gas” throughout the whole manuscript by “uncondensed gas” where it could be misleading, and added $N \ll N_c$ at appropriate places.

What is the red line in Figure 1c, bottom left panel?

13. The red line in the bottom left panel of Fig.1c showed the intensity in one of the interferometer arms, corresponding to the intensity distribution of the condensed photon gas.

We removed the line as it did not provide substantial additional information in this figure.

Why does figure 2b ($N \ll N_c$) have visibility near unity, but figure 4a ($N/N_c = 1.01$) has no more than about 0.3? Surely the visibility should just increase with increasing particle number.

14. Generally, the visibility increases with increasing photon numbers, as the relative influence of technical noise is reduced. However, the data obtained for very low total photon numbers shown in Fig.2 were measured under continuous pumping conditions, which significantly enhanced the maximum measurable visibility due to much longer measurement times. In the condensed regime, continuous pumping is technically unfeasible as it dramatically increases the rate of non-radiative processes in the organic dye molecules.

We emphasized the two different experimental procedures in the main text and methods.

In the supplementary material, section 2, "for the sake of simplicity" you have thrown away something interesting, namely correlations at different times not at $t=r'$. Correlations over time for two different positions can be seen to be non-trivial from the Roumpou et al, PNAS, 2012, showing how correlations propagate. Not only could your calculations presumably be extended to include these correlations, also your data presumably can be re-analysed to extract them. While including this analysis would not be necessary for publication in Nature Communications (in my opinion), it would enhance the importance and interest in your article if you found propagating correlations.

15. By using the inverting retroreflector our experimental data is limited to only show correlations at $r=-r'$. To our understanding, the propagating (e.g. reemerging) correlations are attributed to interactions as well as pump and decay noise, as pointed out in Ref. [31]. Such effects are not included in our theoretical model of the ideal Bose gas. We have checked our experimental data and could not find evidence for propagating correlations.

What is the origin of the numerical coefficients in equation 29 of the supplementary material? Also, in the supplementary material, "Sakurai" is misspelt.

16. The numerical values in equation (29) appear when solving equation (28) for $|g^{(1)}(\tau_c)|=0.5$ and have no deeper physical meaning to our knowledge.

We fixed the spelling of Ref. S1 in the Supplementary information.

Reply to Referee #2:

Referee: This is a nice experiment in many ways, basically a redo of the in-plane ("transverse") coherence measurements of polariton condensates, esp. Ref. 14, for the purely photon condensate. This has the additional feature that the photons are well thermalized, while in many polariton condensates the gas is not fully thermalized.

However, this paper is strangely presented, and feels in some ways like a work in progress rather than a fully developed work.

First, the title and abstract present the main point as measuring the thermal DeBroglie wavelength. But this is not much new. The DeBroglie wavelength is a single-particle property that does not depend on many-body correlations. It can easily be determined (and has been determined) in polariton and photon gas experiments by simple measuring the average k-vector in k-resolved emission spectra. Making this paper mainly about "we can deduce the DeBroglie wavelength through a much more difficult and involuted process" is poor marketing.

This paper is really about the in-plane correlation measurements, and as such should be titled simply "First-order coherence measurements in a 2D..." Then at low density the interesting story is that a theoretical prediction (line 109 on page 5) is confirmed. When I first read the paper it seemed this formula was just pulled out of the air with no justification or citation. But it turns out it is extensively justified in the SI file. The authors need to make this connection clearly by referring to the SI file and giving a brief description of the theory presented there.

More interesting to me is the measurement of the correlation/coherence length in the BEC regime. Nice results for this are given, and are compared to theory in Figure 4c. But once again, on first reading I could not find a citation, discussion, or formula for the dotted and solid curves in Figure 4c (or the curves in 4b). The connection to the SI file (i.e. directly citing the SI file) and a brief summary of the theoretical approach needs to be in the main text.

My conclusion is that this is indeed an interesting paper, but needs major revision. The authors have made it seem as though it is a breakthrough to be able to measure the thermal wavelength (a single-particle property) in the real-space regime rather than the k-space regime. Rather, the main story of this paper needs to be the confirmation of the theory for

correlations, which is presently buried in the SI file.

Our reply: We agree with the Referee that the previous title of the manuscript unintentionally reduced the findings presented in this paper to the de Broglie wavelength of a thermalized photon gas. To account for the measurements of the correlations of the Bose-Einstein condensate, we changed the title to be more generic, as suggested by the Referee. We also substantially revised both the abstract and the introductory part of the main text.

We agree with the referee that a Fourier transform of the k -resolved emission spectra is equivalent to a direct in-plane measurement of the spatial correlations. However, as known from exciton-polariton systems (see a statement by Roumpos et al. [31]), the applicability of Fourier spectroscopy is limited by noise issues and “the direct measurement of $g^{(1)}(x,-x;t)$ is the only way to reliably extract λ_{eff} ”. Certainly, whether the direct in-plane measurement or the measurement of the average k -vector turns out to be more preferable depends on the system under investigation. In our system, the interferometric measurement turns out to be the experimentally straightforward way, and it enabled us to measure for the first time the genuine thermal de Broglie wavelength in a two-dimensional photonic gas.

Following the referee’s advice, we shifted the focus of this manuscript towards spatial correlations of a two-dimensional photon gas in the uncondensed and condensed phase, and the confirmation of the theory of correlations. To this end, we also added a detailed discussion of the correlations in the vicinity of the phase transition. To further clarify the findings given in the manuscript, we included the main ideas and results from the Supplementary information in the main text. We also extended the discussion of the experimental imperfections leading to the shown deviation from the theoretical expectations in Figs. 4b and 4c. With these revisions, we are confident that the suggestions by Referee 2 have significantly improved the quality of this paper.

Reply to Referee #3:

Referee: The manuscript by Damm et al. reports an experimental study of the coherence properties of a Bose gas of photons in a dye-filled optical microcavity. Using a quite standard optical set-up for detection of spatio-temporal coherence, the authors characterize the behaviour of the first-order g_1 coherence function in the two extreme regimes of photon number well below and above the condensation critical point.

I have mixed feelings about this manuscript.

The general idea of this measurement is a quite standard one, and very similar experiments have been carried out with other related systems, in particular polaritons. However, I have to admit that the quality of the experimental data is much cleaner than the one of most polariton experiments which are typically plagued by sample disorder. Here the very smooth data curves make the conclusions very convincing. Some results are novel and typical of photon BEC's, e.g. the measurement of the system-temperature-dependence of the de Broglie wavelength. The overall high quality of the data reinforces the feeling that photon BEC are indeed an excellent work-horse for studies of more sophisticated aspects of long-range phase coherence.

As a most serious flaw of the manuscript, I am afraid that the study is still incomplete as it does not cover all possible regimes around the BEC critical point. From what I remember of the general theory of BEC (see e.g. the 1999-2000 Cohen-Tannoudji's lectures available on the College de France website <http://www.phys.ens.fr/~cct/college-de-france/1999-00/1999-00.htm>), the g_1 has three main regimes as a function of particle density: well below condensation (where it matches the de Broglie wavelength), above condensation (where coherence extends over the whole sample) and an intermediate regime when the system is quantum degenerate and the coherence length is far longer than the de Broglie wavelength, but the system is not yet condensed. This last regime is never explicitly mentioned in this manuscript, which at several points rather gives the potentially misleading impression that BEC occurs as soon as quantum degeneracy is reached. A trace of this behaviour is anyway visible in

fig.4c as the theoretical solid curve start deviating from the classical thermal value already well before the $N/N_c=1$ critical point. For the manuscript to be complete, I think that some discussion of this regime is essential, with some experimental data showing the increased coherence length as an effect of quantum degeneracy before condensation.

In summary, my feeling is that the manuscript is interesting and promising, but it still requires

additional data and additional discussion to be complete and ready for publication. In its present form I can not recommend it for publication, as it might have a misleading effect on the community.

If the authors agree in performing these additional experiments and correspondingly updating the text, I am looking forward to see a revised version of the manuscript for further review.

Our reply: We agree with the Referee that our previous manuscript solely investigated the condensed and (classical) thermal regime, and therefore the study of spatial correlations in the thermalized, two-dimensional photon gas was not complete in a strict sense. We want to thank the Referee for his annotations on the manuscript as it helped to improve the presentation of our findings.

Encouraged by the Referee's remarks, we significantly expanded the discussion of correlations close to the phase transition. To cover this regime with the present data in more detail, we included a discussion of the apparent bimodal decay of correlations into the main text of the manuscript, supported by an additional figure (Fig.5). In short: We analyzed data for total photon numbers slightly above the critical photon number ($N/N_c=1.01$ to 1.08) with respect to the bimodal behavior of the observed decay of correlations in detail and performed additional numerical calculations for the corresponding parameters. In this regime, the lowest transverse modes are evenly occupied (see inset of Fig.5) and the system behaves similar to a homogenous two-dimensional system, showing exponentially decaying correlations accompanied by a Gaussian contribution at short length scales up to the thermal de Broglie length. For higher occupations, e.g. the $N/N_c=1.08$ data trace, the exponential decay constant is larger than the oscillator length ($\sim 7.7\mu\text{m}$), signaling the onset of the regime of a dominantly occupied ground mode.

We point out that besides this intermediate regime mentioned by the referee, in general there also is a further regime even closer to $T=T_c$ expected in the presence of interactions, where according to the Ginzburg-Landau-theory, one expects a universal critical scaling of the order parameters, e.g. the correlation length of the photon gas. We completely agree with the referee that a further study of the regimes near criticality is of large interest. At present, more accurate measurements are however limited in the regime closely below the condensation threshold. Undesired non-radiative properties of the dye medium require us to run the experiment in pulsed mode as we approach the critical particle density. Although these pulses are long (600 ns, i.e. quasi-cw) with respect to the intrinsic photon gas thermalization

timescales (ps to ns), this operational mode limits the maximum exposure time and lowers the signal-to-noise ratio. Correspondingly, the light levels for the uncondensed photon gas are too small to be detected with the required resolution and sensitivity. In addition, relative shot to shot intensity fluctuations of order of 0.2% prevent a more detailed study of the region slightly above the condensate threshold. Therefore, the acquisition of additional experimental data covering these regimes is beyond the scope of the apparatus described in the present manuscript. Note that the measurements performed at very low photon numbers, as shown in Fig.2, were performed at continuous pumping conditions at low powers, which improved the signal-to-noise ratio.

However, the described (quasi) long-range correlations are expected to be examinable in much more detail within box potentials, where no true Bose-Einstein condensate exists, and thus the quantum degenerate regime extends over a broader range of phase-space densities. In a corresponding arrangement, with a signal to noise ratio comparable to the one of the present apparatus, the studies suggested by the referee should straightforwardly be possible, and are planned to be the subject of future works. This is mentioned in the outlook of our manuscript.

In our manuscript, we have besides the addition of Fig.5 also considerably modified the discussion, which now explicitly mentions the intermediate regime and describes our corresponding presently obtained results. Further comments were added to clarify the used cw or pulsed pumping conditions, respectively.

We hope that the revised manuscript version clarifies the concerns of this referee, and are looking forward to further communication on our manuscript.

Summary of changes:

- The manuscript title, abstract and introduction were revised, following the suggestions of referee 2. Also, several following parts of the manuscript were considerably modified, as in more detail described in our response to referee 1.
- We included several main ideas and results previously only given in the Supplementary Information into the main text of the manuscript. We slightly modified the Methods section and updated citations in the Supplementary Information.
- A new figure (Fig.5) has been introduced, see also our response to referee 3, and the intermediate regime where the system is quantum degenerate but not yet condensed is discussed. Also, a more detailed description of the used pumping conditions (cw and pulsed, respectively) was added.
- We introduced subheadings in the manuscript body and the methods section to be conform to the Nature Communication style.
- We added several references and accordingly reordered the bibliography.

Reviewers' comments:

Reviewer #1 (Remarks to the Author):

You seem to have responded appropriately to all of the comments by all the referees, myself included. As such, I reiterate my original recommendation to publish in Nature Communications, this time with only some very minor details that could be amended:

- Line 188: what is the theoretical uncertainty in the thermal de Broglie wavelength?
- A linguistic peeve. I'm not a big fan of the word "bimodal" to describe the decay. The has been taken from the atomic BEC community which misuses it to describe a peak on top of another peak. "Bimodal" means having two separate maxima. Perhaps "compound" decay might be more precise than "bimodal".
- I do not believe the argument on line 251 that the signal to noise for $g(1)$ drops dramatically in going from $N/N_c = 1.01$ (above threshold) to, say 0.98 (below threshold). That said, I also do not see the need for more data just below threshold (as requested by referee #3): the points being made are already proven.
- I am surprised that such low visibilities (about 0.003) are detected in Fig 5, when about 0.1 seems to be the noise floor in Fig 2c. You might like to explain this difference in the main text.

Reviewer #2 (Remarks to the Author):

I have no further objections; the authors have addressed the issues adequately.

Reviewer #3 (Remarks to the Author):

I very much appreciate the constructive reaction of the authors' to Referee reports. The revised version of the manuscript is definitely improved and provides answers to almost all my previous concerns. On this basis I am (almost) ready to recommend publication provided the authors take into account the following remarks:

- pag.4: the expression "the photon gas becomes two-dimensional with an optical dispersion as for a massive particle" might result a bit obscure to unexperienced readers.
- pag.5: when quoting 94000 as the critical photon number, they should specify at which temperature.
- I think it could be useful if the authors clearly mentioned (perhaps already in the abstract, and then on pag.3) that they are dealing with a non-interacting photon gas.
- at bottom of pag.2, references to [25-33] are somehow misleading. The conclusion of [33] is that algebraic order is broken in non-equilibrium 2D systems (unless some strong anisotropy is present). On the other hand, non-equilibrium BKT (on intermediate length scale) was found in Phys. Rev. X 5, 041028 (2015).
- The authors may add to the main text the formula giving the critical photon number as a function of temperature and trap frequency $\sim (T/\omega)^2$ (as given in eq.67 of Hadzibabic-Dalibard), together

with a brief discussion of BEC of non-interacting gas in 2D harmonic trap.

-I remain slightly puzzled by the polarization issues. The authors should mention in the main text what is the mechanism that pins the polarization of the condensate (in the cavity, before the polarizer). Do I understand correctly that for large $N \gg N_c$ the authors are observing a pure condensate concentrated in a single state rather than a fragmented distribution over the two quasi-degenerate polarization states of the ground state? Is there experimental evidence for this?

-I would have expected that the bimodal behaviour discussed at the beginning of pag.11 should be found for N slightly below N_c and not slightly above N_c ; on the other hand, a condensate with long-range order should be found for $N > N_c$. The authors should comment why they are experimentally finding a bimodal distribution at $N/N_c = 1.01$ and 1.02 ? Is it due to some finite-size correction that shifts the exact value of N_c from the thermodynamical limit prediction $\sim (T/\omega)^2$?

Reply to the referees and list of changes

We are pleased about our manuscript being very well received by all three referees. We thank them for their insightful comments, which are addressed in the following.

Correspondence with Referee #1:

Referee: You seem to have responded appropriately to all of the comments by all the referees, myself included. As such, I reiterate my original recommendation to publish in Nature Communications, this time with only some very minor details that could be amended:

- Line 188: what is the theoretical uncertainty in the thermal de Broglie wavelength?

1. Our Reply: The dominant uncertainty for the theoretical value of the thermal de Broglie wavelength stems from the uncertainty of the effective photon mass $m_{\text{eff}} = (n/c_0)^2 hc_0 / \lambda_c$. The refractive index $n(T=297(3)\text{K})=1.431(1)$ of ethylene glycol is well examined as a function of temperature. The uncertainty of the cutoff wavelength $\lambda_c=583.0(5)$ nm is given by the resolution of the spectrometer. We then find the theoretical de Broglie wavelength to be $\lambda_{\text{th}} = 1.482(2)$ μm .

We added the uncertainties of the effective photon mass and the theoretical de Broglie wavelength to the main text.

- A linguistic peeve. I'm not a big fan of the word "bimodal" to describe the decay. This has been taken from the atomic BEC community which misuses it to describe a peak on top of another peak. "Bimodal" means having two separate maxima. Perhaps "compound" decay might be more precise than "bimodal".

2. We completely agree with the referee that the here used meaning of the word "bimodal" is not very precise, though being very frequently used in the AMO community. We have tried to avoid the usage of this word as much as possible.

- I do not believe the argument on line 251 that the signal to noise for $g(1)$ drops dramatically in going from $N/N_c = 1.01$ (above threshold) to, say 0.98 (below threshold). That said, I also do not see the need for more data just below threshold (as requested by referee #3): the points being made are already proven.

3. For these values of N/N_c , a small variation in the total photon number from slightly above to slightly below threshold dramatically impacts the signal to noise in the area of the ground mode, as additional photons upon the onset of Bose-Einstein condensation become concentrated in the trap center, which enhances the signal quality. Given that for N slightly below N_c , to avoid bleaching of the dye, we still have to operate in a pulsed pumping mode, this prevents us from taking reasonable data at the present level of the signal to noise ratio.

- I am surprised that such low visibilities (about 0.003) are detected in Fig. 5, when about 0.1 seems to be the noise floor in Fig. 2c. You might like to explain this difference in the main text.

4. We thank the referee for pointing out the apparent inconsistency. The data shown in Fig. 5 as well as in Fig. 4b already account for a detection noise characteristics given by the camera chip. Thus, the offset of ~ 0.1 visible in Fig. 2c has been subtracted in the figures shown later. Note that this correction for a global offset differs from the position dependent noise correction used to mimic the detector characteristics in the numerical calculations, as explained in the Supplemental Information (Fig. S6).

To be consistent throughout the whole manuscript we applied the background correction also to the data shown in Fig. 2c. We changed the figure and the main text.

Correspondence with Referee #2:

Referee: I have no further objections; the authors have addressed the issues adequately.

Our Reply: We thank the referee for his comments which helped to significantly improve the manuscript.

Correspondence with Referee #3:

Referee: I very much appreciate the constructive reaction of the authors' to Referee reports. The revised version of the manuscript is definitely improved and provides answers to almost all my previous concerns. On this basis I am (almost) ready to recommend publication provided the authors take into account the following remarks:

-pag.4: the expression "the photon gas becomes two-dimensional with an optical dispersion as for a massive particle" might result a bit obscure to unexperienced readers.

1. Our Reply: We expanded the corresponding paragraph to clarify the statement.

-pag.5: when quoting 94000 as the critical photon number, they should specify at which temperature.

2. We added "T=300K (room temperature)" to the according sentence in the main text

-I think it could be useful if the authors clearly mentioned (perhaps already in the abstract, and then on pag.3) that they are dealing with a non-interacting photon gas.

3. We agree with the referee, that one feature of the presented findings is the good agreement with theory for the non-interacting Bose gas. We have added a corresponding remark on page 3 of the revised manuscript.

-at bottom of pag.2, references to [25-33] are somehow misleading. The conclusion of [33] is that algebraic order is broken in non-equilibrium 2D systems (unless some strong anisotropy is present). On the other hand, non-equilibrium BKT (on intermediate length scale) was found in Phys. Rev. X 5, 041028 (2015).

4. We agree that reference [33] demonstrates the absence of algebraic order and thus appears somehow out of line with the other references which deal with the presence of BKT order. However, strictly speaking, also ref. [31] find non-equilibrium aspects in the correlation function, such that we kept citation [33].

As suggested by the referee we added a citation for the quoted paper.

-The authors may add to the main text the formula giving the critical photon number as a function of temperature and trap frequency $\sim(T/\omega)^2$ (as given in eq.67 of Hadzibabic-Dalibard), together with a brief discussion of BEC of non-interacting gas in 2D harmonic trap.

5. We agree with the referee that the addition of the formula for the finite, temperature-dependent critical particle number needed to achieve condensation is helpful. In our manuscript, we expanded the corresponding paragraph on page 5 and added the mentioned formula.

-I remain slightly puzzled by the polarization issues. The authors should mention in the main text what is the mechanism that pins the polarization of the condensate (in the cavity, before the polarizer). Do I understand correctly that for large $N \gg N_c$ the authors are observing a pure condensate concentrated in a single state rather than a fragmented distribution over the two quasi-degenerate polarization states of the ground state? Is there experimental evidence for this?

6. We apologize if our previous answer regarding the polarization of the condensate was misleading or incomplete. To our knowledge there is no mechanism that prefers one polarization state over the other inside the cavity. This is supported by our finding of the good agreement of the measured critical particle number N_c and phase-space-density to achieve condensation with the expectation for a two-fold degenerate gas, as presented in this manuscript. We also have no further experimental evidence that this two-fold degeneration is lifted or changes as a function of condensate fraction. However, when measuring the degree of polarization of the condensate emission outside of the cavity, we find a slight linear polarization of order of 60:40 aspect ratio. As mentioned in reply 8 to referee #1 previously, this could be attributed to stress-induced birefringence inside the microcavity mirror substrates exerted by the used mirror mounts.

-I would have expected that the bimodal behaviour discussed at the beginning of pag.11 should be found for N slightly below N_c and not slightly above N_c ; on the other hand, a condensate with long-range order should be found for $N > N_c$. The authors should comment why they are experimentally finding a bimodal distribution at $N/N_c = 1.01$ and 1.02 ? Is it due to some finite-size correction that shifts the exact value of N_c from the thermodynamical limit prediction $\sim(T/\omega)^2$?

We thank the referee for hinting at the influence of finite-size effects. Indeed, in the thermodynamic limit the bimodal behavior of the correlations is expected to appear only for $N < N_c$, while upon condensation for $N > N_c$ the spatial coherence should become long-range immediately. In other words, in the thermodynamic limit the chemical potential vanishes right at the critical particle number, which implies a divergence of the first-order correlation length.

In our system ($\sim 10^5$ particles) finite-size effects become important. At $N = N_c$ (with N_c as defined in the thermodynamic limit and assuming a noninteracting photon gas) the chemical potential is not strictly clamped to 0, but it still takes finite – although very small – values. This leads to a softening of the phase transition, the correlation length does not discontinuously jump to very large values at N_c but smoothly increases, as N is increased beyond N_c . Correspondingly, the bimodal behavior of the correlations can be observed in the “nominally” condensed phase for not too large values N . We have performed numerical calculations of the correlations for system, which confirm the influence of these finite-size corrections. Despite of this we emphasize that our data still demonstrates the sharp increase of coherence upon the onset of Bose-Einstein condensation.

We have added a comment on the influence of finite-size effects in the manuscript.

Summary of changes:

- As was suggested, we reordered the basic description of Bose-Einstein condensation of photons, and added both the formula for the critical particle number and a discussion on the influence of the finite size effects.
- We offset-corrected the data shown in Fig. 2c to be consistent with Figs. 4b,c and 5, and changed the corresponding parts in the main text (page 6) and the figure caption.
- We added the uncertainty in the predicted effective photon mass and thermal de Broglie wavelength on page 8.
- On page 3, the issue that we compare with ideal Bose gas theory is now mentioned.
- One reference (*Phys. Rev. X* **5**, 041028 (2015)) and the statement of data availability has been added.

REVIEWERS' COMMENTS:

Reviewer #1 (Remarks to the Author):

I consider that this manuscript is suitable for publication in Nature Communications in its present condition.

Reviewer #3 (Remarks to the Author):

I am happy with the authors' revised version. I can finally recommend publication.